# Systematic assessment of obesity-related risk factors in renal cancer etiology: A longitudinal risk and Mendelian randomization analysis

Karine Alcala[1], Daniela Mariosa[1], Sara Jacobson[2], Claudia Coscia-Requena[1], Niki Dimou[1], Oskar Franklin[2], Richard M. Martin[3,4], George Davey Smith[3], Marc J. Gunter[5], Paul Brennan[1], Michael Pollak[6], Ryan Langdon[1], Mattias Johansson[1]*

1 International Agency for Research on Cancer (IARC/WHO), Lyon, France, 2 Department of Diagnostics and Intervention, Surgery, Umeå University, Umeå, Sweden, 3 MRC Integrative Epidemiology Unit (IEU), Bristol Medical School, Department of Population Health Sciences, University of Bristol, Bristol, United Kingdom, 4 National Institute for Health Research (NIHR) Bristol Biomedical Research Centre, University Hospitals Bristol and Weston NHS Foundation Trust and University of Bristol, Bristol, United Kingdom, 5 Cancer Epidemiology and Prevention Research Unit, School of Public Health, Imperial College London, London, United Kingdom, 6 Departments of Oncology and Medicine, McGill University, Montreal, Quebec, Canada

* johanssonm@iarc.who.int

## Abstract

### Background

Excess body adiposity is an established cause of renal cancer, but underlying molecular pathways mediating this relationship remain unclear. This study aimed to systematically evaluate a panel of obesity-related risk factors as potential mediators in renal cancer etiology.

### Methods and findings

We used two complementary approaches to evaluate obesity-related risk factors in renal cancer etiology: (i) direct risk factor assessment in longitudinal cohorts and (ii) genetically proxied risk factors through two-sample Mendelian randomization (MR). Direct risk-factor association-analyses (i.e., cohort analyses) were based on the UK Biobank cohort study (472,337 cohort participants, including 1,382 incident renal cancer cases diagnosed during 5,586,414 person years of follow-up) and the Northern Sweden Health and Disease Study (NSHDS) for fasting insulin (204 pairs of cases and controls, ongoing recruitment and follow-up since 1985). We used Cox proportional hazards regression models to evaluate the association between risk factors and renal cancer risk with adjustment for age, sex, center of recruitment, education, smoking and alcohol drinking status. Two-sample MR analyses were based on a genome-wide association study (GWAS) of renal cancer (27,213 cases, 486,846 controls). We used the inverse-variance weighted (IVW) approach to estimate the

**Data availability statement:** The data underlying the results presented in the study are available from UK Biobank upon request (https://www.ukbiobank.ac.uk) and GWAS summary statistics from https://www.nature.com/articles/s41588-024-01725-7. The code used in the analysis is available from Gitlab: https://gitlab.com/Karine.Alcala/obesity_rcc/-/tree/79b0a4566da2a0a29f3bb-f08eaf1d75f66691f69/ and archived in Zenodo: https://doi.org/10.5281/zenodo.17640162.

**Funding:** MJ and RMM were supported by a Cancer Research UK programme grants: the Obesity-related Cancer Epidemiology Programme (grant number PRCPGM-May25/100001) and The Integrative Epidemiology Programme (C18281/A29019). PB and MJ were supported by World Cancer Research Fund (WCRF UK, IIG_2019_1995). OF was supported by grants from the Swedish Society of Medicine (SLS-960379) and Bengt Ihre Research Fellowship. GDS works within the MRC Integrative Epidemiology Unit at the University of Bristol, which is supported by the Medical Council (MC_UU_00032/1). RMM is a National Institute for Health Research Senior Investigator (NIHR202411). RMM is supported by a Cancer Research UK 25 (C18281/A29019) programme grant (the Integrative Cancer Epidemiology Programme). RMM is also supported by the NIHR Bristol Biomedical Research Centre which is funded by the NIHR (BRC-1215-20011) and is a partnership between University Hospitals Bristol and Weston NHS Foundation Trust and the University of Bristol. SJ was supported by grants from the strategic board for research, Umeå University (FS 2.1.6-59-23) and Svensk-Franska Stiftelsen (F0025_230413). The funding organizations had no role in the design and conduct of the study; collection, management, analysis, and interpretation of the data; and preparation, review, or approval of the manuscript.

**Competing interests:** I have read the journal's policy and the authors of this manuscript have the following competing interests: GDS is an Academic Editor on PLOS Medicine's editorial board. All other authors declare no competing interests.

**Abbreviations:** BMI, body mass index; ccRCC,

association between risk factors and renal cancer risk. Mediation analyses were performed for traits displaying directionally consistent associations with renal cancer risk in both the cohort and MR analyses using the product method. We found consistent positive associations with renal cancer risk for fasting insulin (odds ratio per standard deviation increment [$OR_{MR}$]: 2.24, 95% confidence interval [95% CI]: 1.19, 4.22; $p = 0.01$; hazard ratio per standard deviation increment [$HR_{cohort}$]: 1.43, 95% CI: 1.02, 2.00; $p = 0.04$), triglycerides ($OR_{MR}$: 1.11, 95% CI: 1.05, 1.17; $p < 0.001$, $HR_{cohort}$: 1.23, 95% CI: 1.11, 1.38; $p < 0.001$), diastolic blood pressure (DBP) ($OR_{MR}$: 1.14, 95% CI: 1.04, 1.26; $p < 0.001$, $HR_{cohort}$: 1.11, 95% CI: 1.05, 1.17; $p < 0.001$) and consistent inverse associations with renal cancer risk for sex-hormone binding globulin (SHBG) ($OR_{MR}$: 0.80, 95% CI: 0.70, 0.90; $p < 0.001$, $HR_{cohort}$: 0.67, 95% CI: 0.58, 0.76; $p < 0.001$) and high-density lipoprotein (HDL) cholesterol ($OR_{MR}$: 0.93, 95% CI: 0.88, 0.98; $p < 0.001$, $HR_{cohort}$: 0.72, 95% CI: 0.66, 0.77; $p < 0.001$). The main limitation of this study was that we had limited statistical power to evaluate some risk factors.

### Conclusions

Our study highlights roles for fasting insulin, HDL cholesterol, DBP, triglycerides and SHBG in mediating the relationship between body adiposity and renal cancer risk.

---

### Author summary

#### Why was this study done?

- The importance of excess body adiposity and renal caner etiology is well established, but the underlying mechanisms mediating this relationship are unclear.

#### What did the researchers do and find?

- We used two complementary observational approaches to evaluate 20 potential obesity-related risk factors in relation to renal cancer risk, including

  1. direct risk factor assessment in longitudinal cohorts, and

  2. genetically proxied risk factor assessment through Mendelian randomization (MR) in large genome-wide association studies.

- We found consistent evidence for associations with renal cancer risk for fasting insulin, diastolic blood pressure (DBP), high-density lipoprotein (HDL) cholesterol, triglycerides and sex-hormone binding globulin (SHBG).

#### What do these findings mean?

- Our findings highlighted important roles for fasting insulin, HDL cholesterol, DBP, triglycerides and SHBG in mediating the relationship between body adiposity and renal cancer risk.

clear cell RCC; DBP, diastolic blood pressure; eGFR, estimated glomerular filtration rate; GWAS, genome-wide association study; HbA1c, glycated hemoglobin; HDL, high-density lipoprotein; $HR_{cohort}$, hazard ratio, from cohort analysis; IVW, inverse-variance weighted; MR, Mendelian randomization; NSHDS, Northern Sweden Health and Disease Study; $OR_{MR}$, odds ratio, from MR analysis; pRCC, papillary RCC; RCC, renal cell carcinoma; SHBG, sex-hormone binding globulin; STROBE, Strengthening the Reporting of Observational Studies in Epidemiology; UKB, UK Biobank; VIP, Västerbotten Intervention Study.

- Limitations of our study include a relatively small sample size for the risk analysis of fasting insulin, and an incomplete assessment of sex hormones.

- This study advances our understanding of obesity in renal cancer etiology.

## Introduction

Excess body adiposity—commonly estimated as high body mass index (BMI)—is a well-established cause of renal cancer [1,2]. The proportion of renal cancer cases attributable to high BMI (>25 kg/m$^2$) was estimated to be 17% in 2012 [3], but the underlying biomolecular pathways mediating this association remain unclear [4].

Renal cell carcinoma (RCC) is histologically characterized by its accumulation of glycogen and lipids [5,6], yet its mechanistic link to body fatness remains elusive. Intra-abdominal adipose tissue—previously considered inert—is now recognized as an important endocrine organ with wide-ranging influences in the body, including on hormones, inflammatory biomarkers and lipids, all of which might mediate the obesity link with cancer development [7–11]. There is some evidence from in-vitro and population-based genomics studies implicating insulin in RCC etiology [12,13]. Another important and consistent observation is the 2-fold higher RCC incidence in males versus females [1], suggesting a potential role of sex hormones in RCC [14–17]. There is, however, a paucity of studies evaluating the importance of a wider range of potential factors mediated the well-established link between obesity and RCC development.

The aim of this study was to systematically assess the etiological relevance in RCC etiology for a panel of potential obesity-related mediators. We considered evidence generated by analyzing direct exposure assessment in longitudinal data from a large population cohort, along with genetically proxied exposure assessment in a Mendelian randomization (MR) framework based on genetic data from a large genome-wide association study (GWAS).

## Methods

### Analytical strategy

This study was designed to evaluate the importance of potential obesity-related risk factors in renal cancer etiology, and to estimate the extent to which they may mediate the risk-increasing effect of obesity on renal cancer risk. We considered risk factors previously implicated in the etiology of obesity related cancers, including fasting insulin [13] glucose [18], glycated hemoglobin (HbA1c) [18], estimated glomerular rate (eGFR) [19], blood pressure [13], IGF-1 [20], leptin [21], lipids components [22,23], sex hormones [17] and Interleukin-6 [24,25]. We used two complementary approaches; *i) direct risk factor assessment in longitudinal cohorts* and *ii) genetically proxied risk factors in a large GWAS* (i.e., through MR). These approaches were applied in parallel, initially by **(1) evaluating the association of higher BMI on a set of potential mediators** and **(2) assessing the extent to which the potential mediators were associated with renal cancer risk**. For potential mediators with statistical and directional concordant associations with renal

cancer risk in both the cohort (1) and MR (2) analyses, we finally **(3) estimated the proportion of the BMI-effect on renal cancer risk that they could explain** (Fig 1).

## Study population

### UK Biobank

UK Biobank (UKB) is a prospective cohort that enrolled over 500,000 participants, aged 40–60 years, from 2006 to 2010 in the United Kingdom. Descriptions of enrollment, data collection and details on lifestyle and biomarkers have been described previously [26]. We excluded 27,013 participants diagnosed with cancer (except non-melanoma skin cancer, ICD10: C44) before enrollment. Deaths, incident renal cancer (ICD10: C64) were obtained through data linkage to national cancer and mortality registries. Participants were followed until the first primary malignant cancer, death or end of follow-up (defined as the date of lost to follow-up, last date of cancer diagnosis or death by center) (Details of study population in S1 Appendix). All participants provided written informed consent, and the study protocol was approved by the Northwest Multicenter Research Ethics Committee of the United Kingdom. This study accessed relevant UKB data under application number 97846.

### Northern Sweden Health and Disease Study

The Västerbotten Intervention Study (VIP) is a prospective cohort study within the Northern Sweden Health and Disease Study (NSHDS) that is an ongoing cohort study since 1985 [27]. As of April 2023, VIP included more than 114,000 participants from the general Västerbotten population between 40 and 60 years of age. Participants of VIP with a subsequent renal cancer diagnosis (ICD10: C64) between 1987 and 2018 were identified using the Swedish Cancer Registry. Individuals with a previous diagnosis of malignant disease (except non-melanoma skin cancer) were excluded. Equal numbers of control individuals were randomly selected from the same cohort, matched for sex, age (±6 months), date of sampling (±3 months), and freeze/thaw status of plasma samples (S1 Appendix). This study was approved by the regional ethical committee at Umeå University (2016/384-31) and the Swedish National Review Authority (2020-02179 and 2021-06764-02).

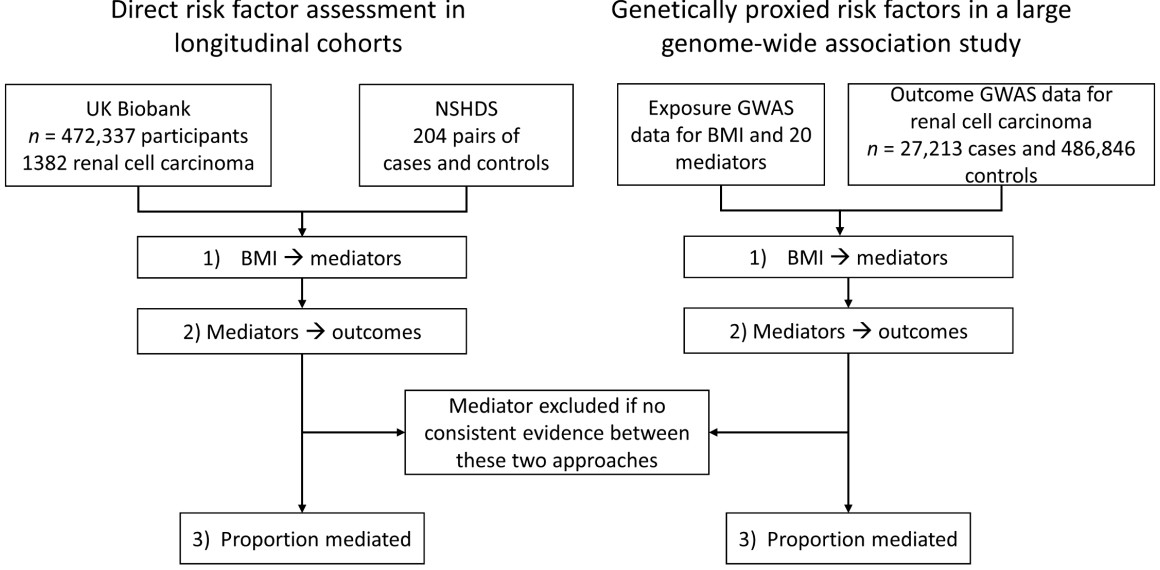

**Fig 1. Flowchart depicts analytical strategy and data sources for (i) directly measured risk factors in longitudinal cohorts, and (ii) genetically proxied risk factors in genome-wide association studies.** NSHDS: Northern Sweden Health and Disease Study. GWAS: Genome-wide association study.

## Genetic instruments

We obtained summary statistics for renal cancer (clear cell RCC [ccRCC], papillary RCC [pRCC], and all types of RCC) from 27,213 cases and 486,846 controls in a multi-ancestry GWAS (mainly European ancestry—24,083 cases and 394,824 controls), excluding individuals in UKB [28]. Summary statistics for all potential risk factors and BMI were obtained from publicly available data (all publication details in Tables A–C in S1 Text). Diastolic and systolic blood pressure GWAS were performed in UKB, based on 375,091 participants of European ancestry [13].

We selected each set of genetic instruments according to the gold standard for MR analyses, previously described (Detailed methods in S1 Appendix) [29]. In brief, for each set of genetic instruments, we excluded non-genome-wide significant SNPs ($p > 5.10^{-8}$) and correlated SNPs with linkage disequilibrium ($r^2 > 0.01$ and separated by less than 10,000kb).

## Statistical analyses

**Cohort analysis.** We initially used linear regression models to estimate the association of BMI on each potential mediator (S1 Fig), and Cox proportional hazards regression models to evaluate the association between BMI, our mediators and renal cancer risk (overall, ccRCC only and pRCC only) [30]. Follow-up time from recruitment to event (diagnosis, death, loss to follow-up, or end of follow-up) was used as the underlying timescale. The end date of follow-up was set as the minimum between the last date of death or cancer diagnosis recorded, by center, from 2020-03-18 for Wrexham to 2022-03-03 for Glasgow. Mediators were transformed for the cohort analyses in accordance with the corresponding GWAS used in our MR analyses (see below, Tables A–C in S1 Text). Each linear regression and Cox model was adjusted for age at baseline, sex (except for sex specific phenotypes), center of recruitment, education, smoking status (never/former/current) and alcohol drinking status (never/former/current). Finally, to calculate the proportion mediated effect, the Cox proportional hazards model of each mediator on renal cancer risk was additionally adjusted for BMI.

For fasting insulin measured in NSHDS, linear regression models were used to investigate the relationship between a standard deviation change in BMI and fasting insulin levels. Odds ratios were calculated using conditional logistic regression models, conditioning on matched case-sets. The natural logarithm of insulin level was used in all analyses. Insulin concentrations below the lower limit of detection (LOD) were replaced with the LOD$\sqrt{2}$.

**Mendelian randomization.** Genetic instrumental variables were identified from GWAS of individuals with European ancestry (Tables A–C in S1 Text) [31,32], adjusted for age, sex and the first 10 principal components. IL-6 and leptin did not have suitable genetic instruments for MR and thus were not further analyzed.

We carried out two-sample two-step MR analysis using the inverse-variance weighted (IVW) approach [31] to estimate the association between *(i)* BMI and renal cancer risk, *(ii)* BMI and each potential mediator, and *(iii)* each potential mediator and renal cancer risk [29,32–34]. Effect estimates were scaled per standard deviation increment in BMI with renal cancer risk and each mediator, and various specific transformation for potential mediators (Tables A–C in S1 Text). Subsequently, we performed multivariable MR analysis to estimate the association for each potential mediator with renal cancer risk with adjustment for BMI. Further details, including the 3 core MR assumptions and procedures for sensitivity analyses, are described in S1 Appendix and S1 Fig.

**Mediation analysis.** Mediation analyses were restricted to potential mediators that displayed directionally consistent associations with renal cancer in both the MR and cohort analyses. Mediation analyses were carried out using the product method [33–35] to estimate the indirect and direct effects for BMI through mediators on renal cancer risk, along with the proportion of the effect of BMI on risk mediated by the mediator (with confidence intervals) [36,37] (S1 Fig).

Statistical analyses were performed using R (version 4.1.2) for UKB and MR and STATA 18 (Stata corp. College Station, TX, USA) for the insulin analyses in NSHDS. This study is reported as per the Strengthening the Reporting of Observational Studies in Epidemiology (STROBE) guidelines (S1 and S2 STROBE Checklist).

## Results

In the longitudinal cohort analysis, we included a total of 472,337 participants from UKB. During 5,586,414 person years of follow-up, 1,382 participants were diagnosed with renal cancer. In NSHDS, we included 204 pairs of renal cancer cases and matched controls. Baseline characteristics of the study participants from the longitudinal cohorts are provided in Tables D–F in S1 Text, and by sex in Table E in S1 Text. The MR analysis was based on GWAS summary statistics generated using 27,213 renal cancer cases and 486,846 controls.

### The association between BMI and potential risk factors

The association between higher BMI and potential mediators, as assessed using both the direct (i.e., cohort analysis: y-axis) and genetically proxied (i.e., MR: x-axis) risk factor assessment, is depicted in Fig 2. The two methods provided consistent evidence for an important role of higher BMI in influencing most assessed risk factors ($r^2$: 0.86). For instance, one standard deviation increase in BMI (1 $SD_{BMI}$) was associated with 0.16 and 0.35 nmol/L higher fasting insulin based

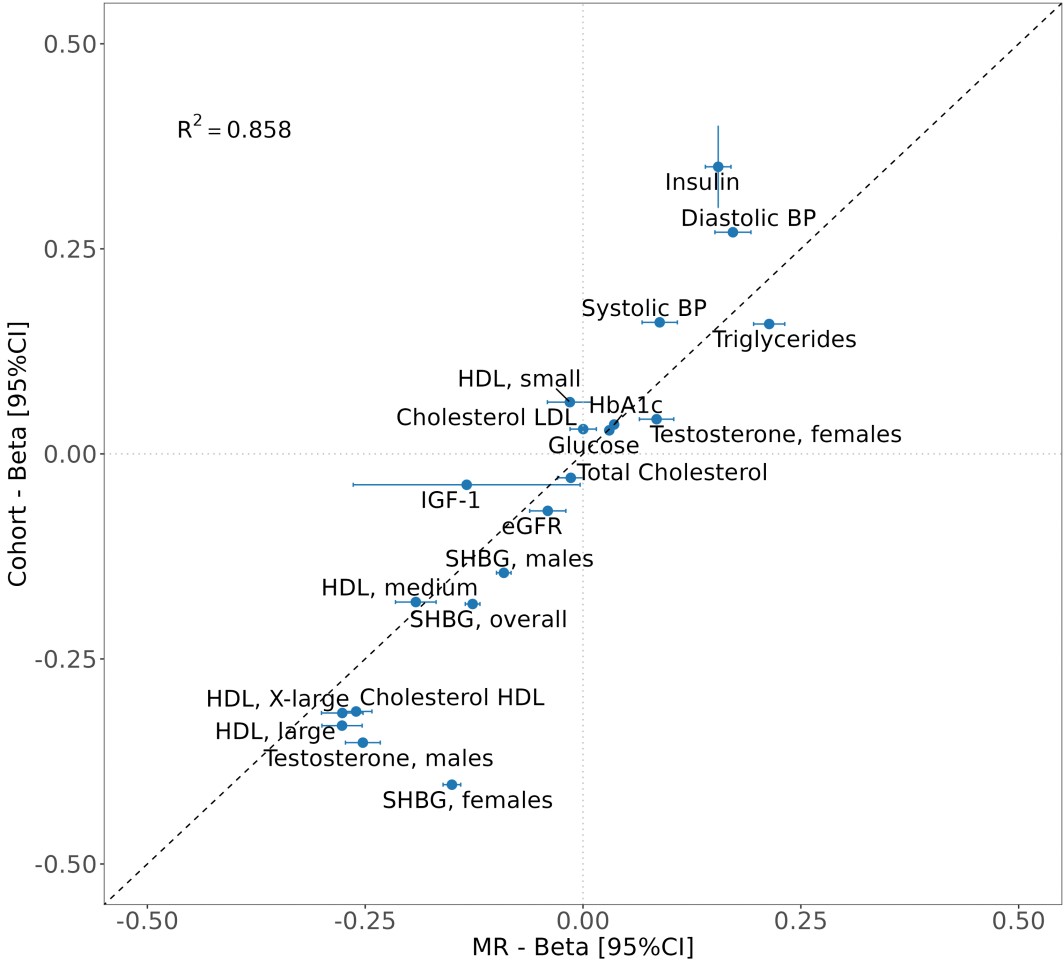

**Fig 2. Figure depicts the association between one standard deviation increment in BMI and potential mediators as measured directly in cohort studies (y-axis) and through genetic Mendelian randomization studies (x-axis).** Error bars indicate 95% confidence intervals for beta estimates (note that most CIs were small in the cohort studies). $R^2$ indicates the correlation between cohort and MR estimates. All linear models were adjusted for age, sex, center of recruitment, education, smoking, and alcohol drinking status.

on the MR and cohort analyses, respectively ($p < 0.001$ in both approaches). Similar positive associations were seen for Glucose, HbA1c, diastolic blood pressure (DBP), systolic blood pressure (SBP), triglycerides, and testosterone in females ($p < 0.001$ in both approaches). Most of the other risk factors were inversely associated with BMI, including sex-hormone binding globulin (SHBG), testosterone in males, high-density lipoprotein (HDL) (medium, large and X-large), IGF-1, eGFR and cholesterol HDL ($p < 0.001$ in both approaches, except for IGF-1 in MR, $p = 0.04$). The only risk factors that did not provide consistent positive or inverse associations with BMI in the cohort and MR analyses were small HDL particles and LDL cholesterol. All effect estimates with confidence intervals are available in Table G in S1 Text.

## Potential risk factor and renal cancer risk

We estimated that a one standard deviation increment in BMI was associated with a renal cancer risk increase of between 33% (HR$_{cohort}$: 1.33, 95% CI: 1.26, 1.39; $p < 0.001$) based on the cohort analysis and 44% based on the MR analysis (OR$_{MR}$: 1.44, 95% CI: 1.35, 1.53; $p < 0.001$) (Table H in S1 Text). Similar associations were observed for BMI with risk of clear-cell renal carcinoma (RCC), but not with risk of pRCC (HR$_{cohort}$: 1.06, 95% CI: 0.83, 1.35; $p = 0.65$, OR$_{MR}$: 1.28, 95% CI: 1.08, 1.51; $p = 0.004$) (Tables I and J in S1 Text).

In risk analyses of potential risk factors (Fig 3), we found statistical and directionally concordant evidence in both the cohort and MR analyses for positive associations with RCC risk for insulin, DBP, and triglycerides. We found corresponding evidence for inverse associations with RCC risk for HDL cholesterol and SHBG. For instance, the MR analysis

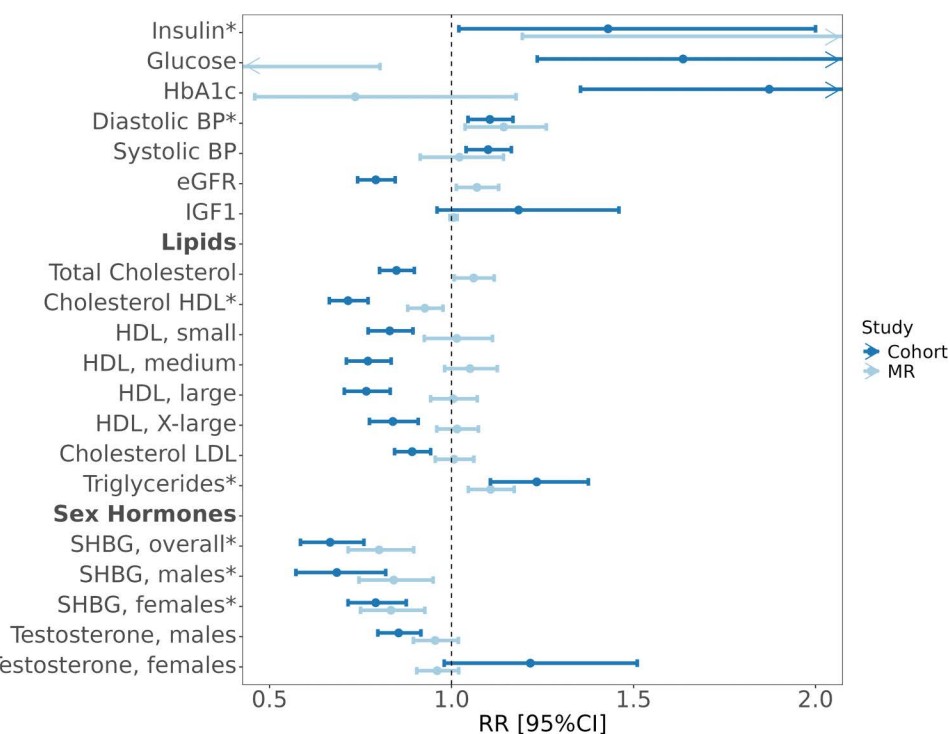

**Fig 3. Forest plot depicts the association between each potential mediator and risk of renal cancer based on MR in light blue and cohort analyses in dark blue.** MR estimated relative risks by calculating odds ratios per standard deviation increment for each potential mediator using the inverse-variance weighted approach. Cohort analyses estimated relative risks by calculating hazard ratios per standard deviation increment for each potential mediator, adjusted for age, sex, education, center of recruitment, smoking and alcohol status, except for fasting insulin which was estimated as odds ratios from conditional logistic regression (the NSHDS sample was a nested case-control study). Potential mediators with evidence for directionally consistent associations with renal cancer risk in both the cohort and MR analyses are denoted with a (*).

suggested that one-standard deviation increment in insulin is associated with a greater than 2-fold increase in RCC risk (OR$_{MR}$: 2.24, 95% CI: 1.19, 4.22; $p=0.01$), with more modest but directionally concordant associations suggested in the cohort analysis (OR$_{cohort}$: 1.43, 95% CI: 1.02, 2.00; $p=0.04$). The associations between triglycerides and RCC risk were modest but similar in the MR and cohort analyses (OR$_{MR}$: 1.11, 95% CI: 1.05, 1.17; $p<0.001$, HR$_{cohort}$: 1.23, 95% CI: 1.11, 1.38; $p<0.001$), with similar results for DBP (OR$_{MR}$: 1.14, 95% CI: 1.04, 1.26; $p<0.001$, HR$_{cohort}$: 1.11, 95% CI: 1.05, 1.17; $p<0.001$). We observed consistent inverse associations with RCC risk for SHBG overall (OR$_{MR}$: 0.80, 95% CI: 0.70, 0.90; $p<0.001$, HR$_{cohort}$: 0.67, 95% CI: 0.58, 0.76; $p<0.001$). We also observed inverse associations with risk for HDL cholesterol (OR$_{MR}$: 0.93, 95% CI: 0.88, 0.98; $p<0.001$, HR$_{cohort}$: 0.72, 95% CI: 0.66, 0.77; $p<0.001$), but found evidence for unbalanced pleiotropy ($p<0.01$) which may have biased the MR-based OR estimate for HDL cholesterol (Table H in S1 Text). We observed discrepancies between the cohort (negatively associated with RCC) and MR analyses (no association) for total cholesterol, HDL particles, LDL and testosterone (Table H in S1 Text). As a sensitivity analysis, for sex-hormones in females, we adjusted our models in the observational analysis with hormone-replacement therapy and menopausal status which did not alter the results (Table K in S1 Text).

## The proportion of the BMI-effect on renal cancer risk explained by each mediator

To estimate the proportion of the BMI effect on renal cancer risk that is mediated by individual risk factors, we carried out a mediation analysis. This analysis was limited to risk factors for which we found consistent evidence for associations with renal cancer risk in both the cohort and MR analyses (Fig 3). Fig 4 graphically depicts the network of relationship connections between elevated BMI, risk factors, and renal cancer risk. This diagram uses information from the analyses

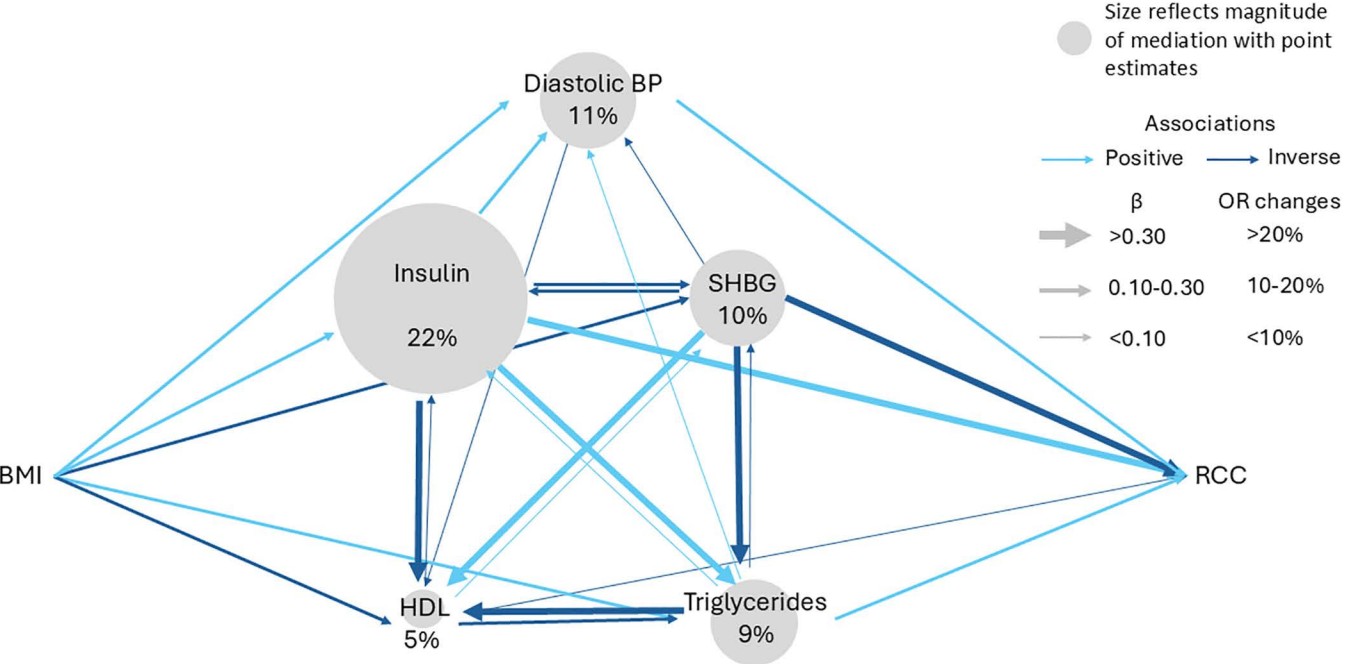

**Fig 4. The diagram summarizes the pairwise associations between BMI, each mediator implicated in renal cancer etiology in this study (i.e., with evidence for association with renal cancer risk in both the MR and cohort-based risk analysis), and renal cancer risk based on univariable MR.** Light blue arrows depict positive associations, and dark blue arrows indicate inverse associations. The thickness of the arrows indicates the strength of association. Continuous associations are expressed as beta estimates, and associations with renal cancer risk are expressed as odds ratios per standard deviation increment. The size of the nodes reflects the amount of the association between BMI and renal cancer risk that is mediated (Table M in S1 Text).

illustrated in Figs 1–3, as well as pair-wise risk factor relations estimated using cohort and MR analyses (Table L in S1 Text). Of note, we found evidence of complex and often bidirectional relations between risk factors.

When considering individual risk factors as mediators, we estimated that insulin mediates between 67% (based on cohort analyses) and 22% (based on MR analyses) of the effect of elevated BMI on renal cancer risk (Fig 4 and Table M in S1 Text). However, we note that the estimated OR between BMI and renal cancer risk in the NSHDS cohort analysis was weaker than the corresponding HR estimate in UKB, and this might have inflated the corresponding mediation estimate for fasting insulin. The corresponding proportion mediated for SHBG was similar between the cohort and MR analysis. HDL cholesterol and triglycerides were estimated to mediate less than 10% of the effect of BMI on renal cancer risk. However, these mediation estimates were based on uncertain indirect effect estimates for the individual risk factors (S2 Fig and Tables M and N in S1 Text). Whereas some evidence was seen in the cohort analysis for indirect effects for fasting insulin (indirect effect: 1.12, 95% CI: 1.04, 1.20), HDL cholesterol (indirect effect: 1.08, 95% CI: 1.06, 1.10), SHBG overall (indirect effect: 1.04, 95% CI: 1.00, 1.08) and SHBG in females (indirect effect: 1.06, 95% CI: 1.02, 1.09), the indirect effect estimates based on the genetic MR analysis had wide confidence intervals (S2 Fig and Tables M and N in S1 Text). We also estimated mutually adjusted association estimates for each putative mediator using both MR and cohort analyses (Table O in S1 Text).

## Discussion

We carried out a systematic assessment of 20 obesity-related risk factors and their association with renal cancer risk by leveraging both direct risk factor assessment in 500,000 longitudinally followed research participants and an MR analysis of 27,213 renal cancer cases with genetic data. Our results confirmed the important influence of elevated BMI on each risk factor, and highlight directionally consistent evidence of association with renal cancer risk for fasting insulin, HDL cholesterol, triglycerides, and SHBG. Mediation analyses suggested roles of these risk factors in mediating the influence of obesity on renal cancer risk.

The etiology of renal cancer is complex and multiple risk factors are thought to contribute. In addition to genetic predisposition and tobacco exposure, excess body adiposity and related risk factors, including hypertension and diabetes, have been considered in renal cancer etiology. We sought to (1) describe the influence of elevated BMI on a broader panel of potential risk factors, (2) investigate their importance in renal cancer etiology, and (3) estimate the extent to which they may mediate the impact of elevated BMI on renal cancer risk. We systematically considered two lines of evidence: *(i)* one based on a traditional approach of direct risk factor assessment in longitudinally followed research participants, and *(ii)* one based on genetically proxied risk factor in a large genetic association study.

(1) The association of elevated BMI with obesity-related risk factors has been well described, and our results were largely consistent with the literature [38–43]. The results suggest that elevated BMI increases concentrations of insulin, blood pressure and triglycerides, but decreases testosterone in men, SHBG and some cholesterol particles. The results were highly concordant in the cohort and MR analyse (Fig 1).

(2) The subsequent risk analysis provided consistent evidence of a risk-increasing effect on renal cancer risk for elevated fasting insulin, DBP and triglycerides, with evidence for risk-decreasing effects of elevated SHBG (Fig 2). These observations have some support from literature, albeit primarily from genetic analyses. The associations with risk for fasting insulin and DBP are consistent with previous reports from our group [13,44], although fasting insulin had not been directly assessed in relation to renal cancer risk in a pre-diagnostic cohort analysis before. *Went and colleagues* recently carried out a phenome-wide MR study across eight common cancers which highlighted inverse associations of SHBG with renal cancer which is consistent with our findings [17]. The inverse association of HDL cholesterol with renal cancer risk is intriguing with a strong inverse association observed with renal cancer risk in the cohort analysis, and a weak inverse association seen in the MR analysis. This observation contrasts with a previous MR study from

*Riscal and colleagues* [45] that suggested a positive association between HDL cholesterol and renal cancer. Our MR analysis used a genetic instrument for HDL with 835 SNPs (compared to 13 SNPs for *Riscal and colleagues*) that was evaluated in relation to renal cancer risk using genetic data from a GWAS of 27,213 cases (instead of 10,784 cases for *Riscal and colleagues*) [45].

We adopted a conservative approach in the mediation analysis by only considering risk factors influenced by (1) elevated BMI (Fig 1) and (2) with evidence for association with renal cancer risk in both the cohort and MR analysis (Fig 2). One previous study has evaluated obesity-related risk factors as potential mediators of the effect of obesity on renal cancer risk. This study estimated that 15% of effect of elevated BMI is mediated by the triglycerides glucose index, an insulin resistance indicator estimated using fasting triglycerides and glucose [46]. This may be compared with our estimate for fasting insulin of 67% in the cohort analysis and 22% in the MR analysis (Fig 4). For the other considered risk factors, we observed mediation estimates of between 3% (for triglycerides) to 27% (for HDL cholesterol) (Table M in S1 Text). However, these analyses were based on one-off measurements of BMI and potential mediators, most of which are likely to vary substantially over time which may result in regression dilution bias [47,48]. We note that the confidence intervals for these mediation estimates were wide, particularly those based on the MR analysis. Whereas many published mediation analyses rely on the point estimates, our study highlight the inherent uncertainty in these analyses and that mediation point estimates should be interpreted with caution. Future studies would benefit from incorporating multiple intra-individual risk factor measurements to assess the extent to which regression dilution may have weakened some of the association estimates in our study, and to improve the precision of the mediation analyses.

When evaluating the directionality of relationships between the identified risk factors, we noted that most risk factors were related and often appeared to influence each other bidirectionally (Fig 4). Indeed, it is not clear whether our results reflect the influence on renal cancer development through one or several pathways. In-vitro studies suggest that insulin inhibits SHBG production and that lower insulin increases SHBG levels [49]. However, other studies suggest that tumor necrosis factor-α [50] or glucose may be involved in SHBG regulation, rather than insulin [51]. In addition, SHBG binds androgens and estrogens, both of which are synthetized from cholesterol [52]. This highlights the need for further studies with a specific focus on sex-hormones metabolism in renal cancer etiology. Taken together, our and previous studies emphasize the complexity of the relationship between obesity and renal cancer etiology, as well as the limitations of the observational approach. Well-designed experimental studies may be required to describe the causal pathways by which obesity influences renal cancer risk in detail.

Our study had important strengths and limitations. We relied heavily on UKB to *(i)* assess the cross-sectional relations between elevated BMI and most of the assessed risk factors, *(ii)* to establish genetic instruments for risk factors of interest, and *(iii)* for the risk analysis based on directly measured risk factors. Whereas UKB is a suitable resource for this line of research, using one, relatively homogeneous and healthy study population may limit the external validity of our findings. The large sample size of the UKB provided statistically robust results but did not solve the potential issue of residual confounding inherent to association analyses based on directly measured risk factors (i.e., the cohort analysis) [53]. An important limitation of our analysis relates to fasting insulin which was not available in UKB, but measured in a separate small case-control study using fasting pre-diagnostic samples. The small sample size available, both for the cohort analysis and for the definition of the genetic instrument, hampered the robustness of the both the MR and cohort analyses for fasting insulin. To circumvent this limitation of traditional observational studies, we additionally used an independent large GWAS of 27,000 renal cancer cases with hundreds of thousands of controls to carry out complementary risk analyses along the lines of two-sample MR. By adopting a conservative approach when interpreting the results in focusing on risk factors with consistent associations with renal cancer risk in both the cohort and MR analyses, the specific risk factors implicated in renal cancer etiology by our study would seem robust.

In conclusion, we evaluated a comprehensive panel of obesity-related risk factors in relation to renal cancer risk using two complementary approaches. We quantified the important influence of obesity on each risk factor and found robust

evidence for a role in renal cancer etiology for fasting insulin, HDL cholesterol, SHBG, DBP, and triglycerides. Our results reinforce prior evidence that insulin is an important causal factor for RCC, and is likely to represent an important link between obesity and RCC development. Our study also highlighted likely important roles for cholesterol and sex steroid metabolism in renal cancer etiology.

## Supporting information

**S1 Fig. Flowchart of the mediation analysis.** NSHDS: The Northern Sweden Health and Disease Study. GWAS: Genome-wide association study. SNPs: Single nucleotide polymorphism. Exp: exposure. Med: potential mediator. Out: outcome. MR: Mendelian Randomization. IVW: Inverse Variance Weighted.
(DOCX)

**S2 Fig. Indirect effect estimates for BMI in the mediation analysis.** Indirect effect estimates for BMI through each risk factor with renal cancer risk as calculated with the product method, confidence intervals estimated through the Sobel method. MR: Mendelian Randomization. BP: Blood pressure. HDL: High-density lipoprotein. SHBG: Sex-hormone binding globulin. CI: Confidence Interval.
(DOCX)

**S1 Text. Table A.** Potential obesity-related risk factors for renal cell carcinomas, in the Mendelian Randomization analyses. *Removed due to missingness in the outcome SNPs, outliers identified by MR-PRESSO or removed by steiger filtering. SNP: single-nucleotide polymorphism. GWAS: Genome-wide association study. INTR: Inverse normal transformation of rank. HbA1c: glycated hemoglobin. IGF-1: Insulin-like growth factor-1. eGFR: estimated glomerular filtration rate. HDL/LDL: High/Low-density lipoproteins. SHBG: Sex-hormone binding globulin. **Table B. Potential obesity-related risk factors for clear cell renal cell carcinomas, in the Mendelian randomization analyses.** *Removed due to missingness in the outcome SNPs, outliers identified by MR-PRESSO or removed by steiger filtering. SNP: single-nucleotide polymorphism. GWAS: Genome-wide association study. INTR: Inverse normal transformation of rank. HbA1c: glycated hemoglobin. IGF-1: Insulin-like growth factor-1. eGFR: estimated glomerular filtration rate. HDL/LDL: High/Low-density lipoproteins. SHBG: Sex-hormone binding globulin. **Table C. Potential obesity-related risk factors for papillary renal cell carcinoma, in the Mendelian Randomization analysis.** *Removed due to missingness in the outcome SNPs, outliers identified by MR-PRESSO or removed by steiger filtering. SNP: single-nucleotide polymorphism. GWAS: Genome-wide association study. INTR: Inverse normal transformation of rank. HbA1c: glycated hemoglobin. IGF-1: Insulin-like growth factor-1. eGFR: estimated glomerular filtration rate. HDL/LDL: High/Low-density lipoproteins. SHBG: Sex-hormone binding globulin. **Table D. Characteristics of the UKB population, by renal cell carcinomas subtypes.** [1]n (%); Median (IQR). RCC: Renal cell carcinoma. ccRCC: clear cell renal cell carcinoma. pRCC: papillary renal cell carcinoma. HbA1c: glycated hemoglobin. IGF-1: Insulin-like growth factor-1. eGFR: estimated glomerular filtration rate. HDL/LDL: High/Low-density lipoproteins. SHBG: Sex-hormone binding globulin. **Table E. Characteristics of the UKB population, by sex.** [1]n (%); Median (IQR). HbA1c: glycated hemoglobin. IGF-1: Insulin-like growth factor-1. eGFR: estimated glomerular filtration rate. HDL/LDL: High/Low-density lipoproteins. SHBG: Sex-hormone binding globulin. **Table F. Characteristics of the Northern Sweden Health and Disease Study (NSHDS) population.** [1]n (%); Mean (95% confidence interval). **Table G. Association of BMI with each potential mediator.** Each cox proportional hazard model was adjusted for age, sex, center of recruitment, education, smoking and alcohol drinking status. *log-transformed, **inverse-normal transformation of rank and Z-score, ***Z-score transformation, †standardized and 1-unit increase in Estradiol. Fasting insulin association with BMI was assess in NSHDS. HbA1c: glycated hemoglobin. IGF-1: Insulin-like growth factor-1. eGFR: estimated glomerular filtration rate. HDL/LDL: High/Low-density lipoproteins. SHBG: Sex-hormone binding globulin. **Table H. Association of potential obesity-related risk factors with risk of renal cell carcinoma using Mendelian randomization and prospective cohort analyses.** Adjustments for the cox models: Age, sex, center, education, alcohol

status and smoking status, • SNPs removed by MR-PRESSO and steiger filtering, *log-transformed, **inverse-normal transformation of rank and Z-score, ***Z-score transformation, †standardized and 1-unit increase in Estradiol. Fasting insulin association with RCC was assess in NSHDS.ᵃ Fasting insulin effect as OR [95% CI]. HbA1c: glycated hemoglobin. IGF-1: Insulin-like growth factor-1. eGFR: estimated glomerular filtration rate. HDL/LDL: High/Low-density lipoproteins. SHBG: Sex-hormone binding globulin. **Table I. Association of potential obesity-related risk factors with risk of clear cell renal cell carcinoma using Mendelian randomization and prospective cohort analyses.** Adjustments for the cox models: Age, sex, center, education, alcohol status and smoking status, • SNPs removed by MR-PRESSO and steiger filtering, *log-transformed, **inverse-normal transformation of rank and Z-score, ***Z-score transformation, †standardized and 1-unit increase in Estradiol. Fasting insulin association with RCC was assess in NSHDS. ᵃ Fasting insulin effect as OR [95% CI]. HbA1c: glycated hemoglobin. IGF-1: Insulin-like growth factor-1. eGFR: estimated glomerular filtration rate. HDL/LDL: High/Low-density lipoproteins. SHBG: Sex-hormone binding globulin. **Table J. Association of potential obesity-related risk factors with risk of papillary renal cell carcinoma using Mendelian randomization and prospective cohort analyses.** Adjustments for the cox models: Age, sex, center, education, alcohol status and smoking status, • SNPs removed by MR-PRESSO and steiger filtering, *log-transformed, **inverse-normal transformation of rank and Z-score, ***Z-score transformation, †standardized and 1-unit increase in Estradiol. Fasting insulin association with RCC was assess in NSHDS. ᵃ Fasting insulin effect as OR [95% CI]. HbA1c: glycated hemoglobin. IGF-1: Insulin-like growth factor-1. eGFR: estimated glomerular filtration rate. HDL/LDL: High/Low-density lipoproteins. SHBG: Sex-hormone binding globulin. **Table K. Comparison between the adjusted and the non-adjusted model for specific female factors.** *Cox proportional hazard model adjusted for age, center of recruitment, education, smoking status and alcohol drinking status. ** Cox proportional hazard model additionally adjusted for Hormone replacement therapy (Yes/no) and menopausal status (Yes/no). RCC: renal cell carcinoma. ccRCC: clear cell renal cell carcinoma. SHBG: Sex-hormone binding globulin. **Table L. Beta estimates between potential mediators.** Association between each potential mediator, bidirectionally, in cohort and MR analyses. BP: blood pressure. SHBG: Sex-hormone binding globulin. HDL: High-density lipoprotein cholesterol. Each model, in the cohort analyses, was adjusted for age, sex, center of recruitment, education, alcohol and smoking status. **Table M. Proportion of BMI effect on renal cell carcinoma mediated.** Each cox proportional hazard model was adjusted for age, sex, center of recruitment, education, smoking and alcohol drinking status. ᵃFasting insulin effect as OR [95% CI]. BP: blood pressure. SHBG: Sex-hormone binding globulin. HDL: High-density lipoprotein cholesterol. **Table N. Proportion of BMI effect on ccRCC mediated.** Each cox proportional hazard model was adjusted for age, sex, center of recruitment, education, smoking and alcohol drinking status. ᵃ Fasting insulin effect as OR [95% CI]. SHBG: Sex-hormone binding globulin. HDL: High-density lipoprotein cholesterol. **Table O. Multivariable analyses of mediators on renal cell carcinomas.** Multivariable analyses between each potential mediator adjusted for each other mediator. Each cox proportional hazard model was adjusted for age, sex, center of recruitment, education, smoking and alcohol drinking status. Not enough SNPs in Insulin instruments to run multivariable MR with another mediator. - Not calculated to due lack of power ($F$-statistic$_{conditional}$ <10). BP: blood pressure. SHBG: Sex-hormone binding globulin. HDL: High-density lipoprotein. (DOCX)

**S1 STROBE Checklist. Checklist of items that should be included in reports of observational studies.** Licensed under CC BY 4.0. Checklist available from https://www.strobe-statement.org/checklists/.
(DOC)

**S2 STROBE Checklist. STROBE-MR checklist of recommended items to address in reports of Mendelian randomization studies.** Checklist available from https://www.strobe-mr.org/.
(DOCX)

**S1 Appendix. Supplementary methods and materials.**
(DOCX)

## Acknowledgments

Data on glycemic traits were generated by MAGIC investigators and downloaded from https://www.magicinvestigators.org

All participants provided written informed consent, and the study protocol was approved by the Northwest Multicenter Research Ethics Committee of the United Kingdom. This study accessed relevant UK Biobank data under application number 97846.

IARC disclaimer:

Where authors are identified as personnel of the International Agency for Research on Cancer/World Health Organization, the authors alone are responsible for the views expressed in this article and they do not necessarily represent the decisions, policy, or views of the International Agency for Research on Cancer/World Health Organization.

Department of Health and Social Care disclaimer:

The views expressed are those of the author(s) and not necessarily those of the NHS, the NIHR or the Department of Health and Social Care.

## Author contributions

**Conceptualization:** Karine Alcala, Daniela Mariosa, Mattias Johansson.

**Data curation:** Karine Alcala, Daniela Mariosa, Sara Jacobson, Ryan Langdon.

**Formal analysis:** Karine Alcala, Daniela Mariosa, Sara Jacobson, Ryan Langdon.

**Funding acquisition:** Richard M. Martin, Paul Brennan, Mattias Johansson.

**Methodology:** Karine Alcala, Daniela Mariosa, Ryan Langdon, Mattias Johansson.

**Resources:** Michael Pollak, Mattias Johansson.

**Supervision:** Ryan Langdon, Mattias Johansson.

**Validation:** Ryan Langdon.

**Visualization:** Karine Alcala.

**Writing – original draft:** Karine Alcala, Daniela Mariosa, Sara Jacobson, Mattias Johansson.

**Writing – review & editing:** Karine Alcala, Daniela Mariosa, Sara Jacobson, Claudia Coscia-Requena, Niki Dimou, Oskar Franklin, Richard M. Martin, George Davey Smith, Marc J. Gunter, Paul Brennan, Michael Pollak, Ryan Langdon, Mattias Johansson.

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
