## [Editor Report · Decision Letter 0]

7 Jul 2025

Dear Dr Johansson,

Thank you for submitting your manuscript entitled "Obesity and renal cancer etiology: a systematic assessment of potential mediators" for consideration by PLOS Medicine.

Your manuscript has now been evaluated by the PLOS Medicine editorial staff and I am writing to let you know that we would like to send your submission out for external peer review.

For clinical studies, please upload a copy of your trial study protocol as a supporting information file. The study protocol should be the version submitted for approval to the institutional review board or ethics committee, should include any amendments to the study protocol, as well as the date of their approval by the institutional review or ethics committee. Please also detail any deviations from the study protocol in the Methods section of your manuscript. The editors will consider the protocol and study conduct prior to a final decision for external review.

Please re-submit your manuscript within two working days, i.e. by Jul 09 2025.

Feel free to email me at atosun@plos.org or us at plosmedicine@plos.org if you have any queries relating to your submission.

Kind regards,

Alexandra Tosun, PhD

Senior Editor

PLOS Medicine

---

## [Decision Letter · Decision Letter 1]

24 Oct 2025

Dear Dr Johansson,

Many thanks for submitting your manuscript "Obesity and renal cancer etiology: a systematic assessment of potential mediators" (PMEDICINE-D-25-02391R1) to PLOS Medicine. The paper has been reviewed by subject experts and a statistician; their comments are included below and can also be accessed here: [LINK]

As you will see, while the reviewers find the study to be generally well-conducted, they express concerns about the conceptual framework as well as its novelty. After discussing the paper with the editorial team and an academic editor with relevant expertise, I'm pleased to invite you to revise the paper in response to the reviewers' comments. We plan to send the revised paper to some or all of the original reviewers, and we cannot provide any guarantees at this stage regarding publication.

We ask that you submit your revision by Nov 14 2025. However, if this deadline is not feasible, please contact me by email, and we can discuss a suitable alternative.

Don't hesitate to contact me directly with any questions (atosun@plos.org).

Best regards,

Alexandra

Alexandra Tosun, PhD

Senior Editor

PLOS Medicine

atosun@plos.org

Comments from the academic editor:

This is an interesting and well-done manuscript that helps characterize why obesity may relate to kidney cancer risk. The study is well-powered and rather well-done but the analytes being examined and the methodology employed are not especially novel. The results are important but not groundbreaking. The study has a key limitation in that the most important result--for fasting insulin--was available only in a subset. This makes it harder to use multivariable models and introduces questions about whether the result is stronger simply because of the use of a different dataset.

I think the manuscript is most interesting from a broader perspective--that of a field transitioning from mostly questionnaire-based methods to intensively-biological methods, and currently grappling with how to interpret such dense data and how it can be used to complement traditional hypotheses.

Specific comments are below:

1. Would the authors consider including metabolomics data from the UK Biobank (and/or NHSDS) to add novelty?

2. The lack of fasting insulin in Biobank is a limitation that should be more forthrightly acknowledged.

3. Would the authors consider examining HBA1c and fasting glucose from the UK Biobank?

4. The network depiction (Figure 4) should be described in the Methods and in much greater detail. Is this a Gaussian Graphical Model, with conditional correlations, or are the lines unconditional? Are lines based on observed levels or MR levels? Which is which? How about the circles--observed or MR? Are results from UK Biobank or NSHDS? Perhaps % mediation can be included in the circle, because the size of circles in the key are misleading. It's a good figure but it needs more footnoting and annotating to be clear.

5. For the insulin result, I think it's important to clarify in the results that the BMI-RCC association was weaker in NSHDS than UK Biobank to begin with, which may (partly) explain the higher percent of mediation for insulin.

6. I would omit most or perhaps all instances of bolding in the discussion.

7. Discussion: regarding "regression dilution bias" in previous papers. Did the current paper address this is some way? This point did not appear to support the strength of the current study?

Comments from the reviewers:

Reviewer #1: This study by Alcala et al is a carefully designed analysis of making the most of observational and Mendelian randomisation estimates to elucidate potential pathways underlying the association of BMI with renal cancer through mediation analyses. Parts of these analyses have been previously explored and some results have been established. The authors were very clear on which parts these were, supported by the relevant literature and which parts needed more evidence, where they contributed. From the potential mediators they investigated, they found that fasting insulin, diastolic BP, triglycerides, sex hormone binding globulin and HDL-C mediated to some extent this association.

I thought this was a very relevant research question and the study design was carefully thought through utilising many data sources to make as robust a conclusion as possible. I found the analytical decisions reasonable. I mainly have minor comments related to the presentation of the results.

Overall, throughout the text, tables and figures, it should be made clear that any results from insulin came from a conditional logistic regression analysis (based on my understanding). First, NSHDS is not mentioned at all in the abstract. Despite any word limit, I think there should be an effort to make this clear in a concise way. Second, throughout including tables/figures, the results for insulin from the observational analyses are presented as HR. This is not quite right, so for insulin it needs to be changed to OR. When insulin is presented along with other risk factors in the same table, the title should be something that covers this too .e.g HR/OR [95% CI] with a footnote explaining that for insulin is OR or Effect size [95% CI] with a footnote.

Line 72: Add "." After "RCC".

Lines 104-105: Could you mention that latest censoring date, if the participant did not die, get cancer or was lost to follow-up?

Line 125: The citation number jumps from 26 to 30, so this needs adjustment and changing the order of references.

Lines 133-135: Wouldn't it be relevant for the Cox models to be adjusted for a SES variable e.g. education or deprivation index? Was there any reason in particular that they were not while the conditional logistic regression analyses were (as per the supplementary figure 1)?

Lines 148-149: This is not very clearly written. Would the following rephrasing be accurate "to estimate the association between i) BMI and each potential mediators, ii) each potential mediator and renal cancer risk, and iii) BMI and renal cancer risk."?

Lines 151-153: Since BMI was the exposure of interest, in the multivariable MR, why didn't you frame it as adjustment of the BMI and renal cancer risk association for each potential mediator and then directly estimating from this the direct effect? Then the indirect effect for each potential mediator would be total effect-direct effect. Are the two approaches equivalent or is there any particular reason you did the analysis this way?

Line 156: Replace "was" with "were".

Lines 161-162: Could you rewrite this as "Statistical analyses were performed using R (version 4.1.2) for UKB and MR and STATA 18 (Stata corp. College Station, TX, USA) for the insulin analyses in NSHDS."

Line 189: According to the Supplementary Table 7, this estimate is 1.32 [1.26, 1.39]. Please double-check and correct as appropriate.

Line 199: Similarly, the OR should be HR.

Line 200: In Supplementary Table 7, the 95% CI for the MR are 1.04-1.17. Please double-check and correct accordingly.

Line 202: Similarly, check the upper confidence bound of the HR (0.76 vs 0.75).

Line 204: Aren't these results still refer to Supplementary Table 7 (rather than 8)?

Line 221: Add an "s" after "depict".

Line 284: Add "d" after "evaluate".

Line 287: Add "d" after "influence".

Line 307: Add "d" after "involve".

References 32 and 34 are the same.

Figure 1: Maybe to make this figure more accurate, split steps 1 and 2 into two separate boxes and then make the arrows that go towards the central box of the "Mediator excluded if…" come from the box of step 2 only (Mediators -> outcomes), as this is what you did from my understanding.

Figure 2: All the vertical 95% CIs apart from insulin are missing; could you add them? In addition, it may make the figure clearer if you remove the background grey lines and instead you add a grey line at 45 degrees.

Figure 3: For the insulin CIs, could you put an arrow-head on the right-hand side to show that the lines continue outside the x-axis limits?

Figures 3 and 5: As suggested for figure 2, please remove the grey background lines to make the figures cleaner.

Figure 4: You may want to change the colours of the arrows to make them more colour-blind friendly.

Supplementary methods:

1. The first sentence of the third paragraph looks like it was left from a previous version that it may have been in the main text. Since you talk about SBP and DBP in the previous paragraph, maybe replace this sentence with just a description of the BMI measurement.

2. When you talk about missing biomarker measurements, could you first mention what was the percent missingness and then say "… of which most (80%) occurred due to …".

3. Mendelian randomisation, 5th paragraph, 3rd line: add "d" after "remove".

Supplementary table 7: In the title you mention kidney cancer. Could you keep it to renal cell carcinoma throughout for consistency and to avoid any confusion among the readers?

Reviewer #2: In this paper, the authors applied regression analysis to the UK Biobank Cohort and Swedish cohort as well as MR analysis to GWAS Consortium data to assess the causal relationship of obesity with renal cell cancer (RCC) including clear cell RCC (ccRCC) and papillary cell RCC (pcRCC). They further explored the relationships of putative mediators including insulin, lipid parameteres, testosterone and SHBG along with their explained variance. They compared the effect size and explained variances of these 2 analyses and found similar directions albeit with substantial differences for some biomarkers. In a network analysis, they found bidirectional relationships amongst these biomarkers in their mediation of the causal association between BMI and RCC.

While the researchers have performed extensive and robust analyses using these large datasets, the main concern relates to the hypothesis and the conceptual framework underlying these analyses. The risk associations between obesity and cancer including RCC are well documented in cross sectional and prospective studies supported by biological plausibility. The latter include abnormal cell signalling due to inflammation, oxidative stress, glucolipotoxicity amongst others. Prospective studies and more recently RCTs with anti-obesity medications have also demonstrated the benefits of weight reduction on cancer risk. These consistent data support the causal role of obesity and cancer events.

Although this analysis may add insights into the mediators for such causal relationships, it has not addressed the intriging question whether this obesity-cancer link is true for all cancer with additional factors modifying the risk of site-specific cancer. As eluded by the authors, there is a male prepondrance for RCC and given the divergent relationships between testosterone and RCC in men and women, sex-specific analysis is needed to improve clarity. Besides, the range of testosterone levels in men and women are markedly different making sex-specific analysis imperative. Some researchers also used different assays to detect the low testosterone levels in women and it is not sure how these data were handled in these large databases with considerable heterogeneity in terms of assays and definitions. A table comparing differences between men and women in the UK and Swedish cohorts will be informative, in particular the menopausal states of women and stages of CKD (see later). There are also sex differences in distribution of anthropmetric indexes and analysis of visceral obesity indexes (e.g. waist circumference) will strengthen the results. Likewise, alcohol and tobacco use are different between men and women and thus, the mediators may be different even if the causal relationship between obesity and RCC holds true in both sexes.

Importantly, the confounding or mediating effects of hyperglycemia in obesity-cancer relationship needs to be clarified. There are studies supporting the association between obesity and hyperglycemia independent of obesity, again supported by biological plausibility. The relationship between insulin and cancer may also be different in people with or without diabetes. Likewise, CKD may lie on the causal pathway leading to RCC with low eGFR being associated with low SHBG. The authors need to take these biological factors into considersation when performing these statistical analyses .

The author needs to provide more explanation on the methdology of the network analysis and propose some biological explanation to these bidirectional relationships. General (BMI) or visceral (waist circumfernce) obesity is due to perturbation of energy metabolism in part due to abnormal insulin secretion and action which in turn can affect lipid and glucose metabolism, vascular biology, inflammation, and other hormonal pathways including SHBG and sex hormones. These biological changes can lead to complications such as CKD which can feedback onto these complex loops. These complexies may not be fully addressed by MR analysis even using multiple statistical tools such as horizontal plieotrophy. and that consistent evidence from different study design is probably more important.

Please define ccRCC and pcRCC in the introduction when they first appeared. Please clarify whether the RCC cases in the UK biobankwere cross-sectionally or prospectively ascertained. Likewise, was the case-control cohort of RCC in the Swedish cohort cross sectional or prospective? If the former, then reporting patient-years can be misleading.

In the supplementary text (p3), please clarify the statement: We additionally excluded correlated SNPs in linkage disequilibrium (r2>0.01 and separated by less than 10 000 kb). Is r2>0.01 correct?

Reviewer #3: The paper considers the relationship between BMI and renal cancer, through both observational analysis and using Mendelian randomization with genetic data. Potential mediators of this relationship are identified, and the proportion of the association between BMI and renal cancer risk mediated by these factors is studied. My comments are as follows.

Some of the causal language may be tempered, for example in line 88 it is said the study is evaluating the "impact" of higher BMI on potential mediators, and in line 128 "the effect" of BMI. Whereas the effects measured, particularly in the cohort analysis, are associational rather than causal.

By labelling the potential mediators as such, it is implicit that these factors sit on the causal pathway between BMI and RCC. Bidirectional analyses were performed among the mediators, but not between the mediators and BMI. Is there any evidence of bidirectional relationships between BMI and the mediators?

The MR-Egger intercept test suggests that directional pleiotropy is present for a number of the mediators, as shown in Supplementary tables 7, 8, and 9. E.g., BMI, HDL, triglycerides, SHBG all have low p-values from the intercept test, suggesting the SNPs may not meet the IV assumptions and the MR estimates may be biased. Has this been considered in the subsequent mediation analysis?

---

* Please upload any figures associated with your paper as individual TIF or EPS files with 300dpi resolution at resubmission; please read our figure guidelines for more information on our requirements: http://journals.plos.org/plosmedicine/s/figures. While revising your submission, we strongly recommend that you use PLOS's NAAS tool (https://ngplosjournals.pagemajik.ai/artanalysis) to test your figure files. NAAS can convert your figure files to the TIFF file type and meet basic requirements (such as print size, resolution), or provide you with a report on issues that do not meet our requirements and that NAAS cannot fix.

After uploading your figures to PLOS's NAAS tool - https://ngplosjournals.pagemajik.ai/artanalysis, NAAS will process the files provided and display the results in the "Uploaded Files" section of the page as the processing is complete.

If the uploaded figures meet our requirements (or NAAS is able to fix the files to meet our requirements), the figure will be marked as "fixed" above. If NAAS is unable to fix the files, a red "failed" label will appear above.

When NAAS has confirmed that the figure files meet our requirements, please download the file via the download option, and include these NAAS processed figure files when submitting your revised manuscript.

* Please provide the ethical approval number and ensure to include the ethics statement in the Methods section.

FIGURES AND TABLES

SUPPLEMENTARY MATERIAL

REFERENCES

STUDY TYPE-SPECIFIC REQUESTS

OBSERVATIONAL STUDIES

* Abstract: Please include the study design, population and setting, number of participants, years during which the study took place (enrollment and follow up), length of follow up, and main outcome measures.

* Please ensure that the study is reported according to the STROBE (or appropriate STOBE extension) guideline (available from: https://www.equator-network.org/reporting-guidelines/strobe) and include the completed STROBE (or STROBE extension) checklist as Supporting Information. Please add the following statement, or similar, to the Methods: "This study is reported as per the Strengthening the Reporting of Observational Studies in Epidemiology (STROBE) guideline (S1 Checklist)." When completing the checklist, please use section and paragraph numbers, rather than page numbers.

* In the manuscript text, please indicate: (1) the specific hypotheses you intended to test, (2) the analytical methods by which you planned to test them, (3) the analyses you actually performed, and (4) when reported analyses differ from those that were planned, transparent explanations for differences that affect the reliability of the study's results. If a reported analysis was performed based on an interesting but unanticipated pattern in the data, please be clear that the analysis was data driven.

* Please state in the Methods section whether the study had a prospective protocol or analysis plan. If a prospective analysis plan (from your funding proposal, IRB or other ethics committee submission, study protocol, or other planning document written before analyzing the data) was used in designing the study, please include the relevant document(s) with your revised manuscript as a Supporting Information file to be published alongside your study and cite it in the Methods section. A legend for this file should be included at the end of your manuscript. If no such document exists, please make sure that the Methods section transparently describes when analyses were planned, and when/why any data-driven changes to analyses took place. Changes in the analysis, including those made in response to peer review comments, should be identified as such in the Methods section of the paper, with rationale.

MENDELIAN RANDOMIZATION STUDIES

* Please ensure that the study is reported according to the STROBE-MR guideline (https://www.equator-network.org/reporting-guidelines/strobe/) and include the completed STROBE-MR checklist as Supporting Information. Please add the following statement, or similar, to the Methods: "This study is reported as per the Strengthening the Reporting of Observational Studies in Epidemiology (STROBE) guideline, specific for mendelian randomization (S1 Checklist)." When completing the checklist, please use section and paragraph numbers, rather than page numbers.

* In the Introduction, please describe the exposure and the evidence for a potential causal relationship between exposure and outcome.

* In the Methods, please explicitly state the 3 core instrumental variable assumptions for the main analysis (relevance, independence, and exclusion restriction), as well assumptions for any additional or sensitivity analysis.

* In the Methods, please describe the MR estimator (e.g., 2-stage least squares, Wald ratio) and related statistics. Detail the included covariates and, in case of 2-sample MR, whether the same covariate set was used for adjustment in the 2 samples.

* If you are presenting an instrumental variable estimate, please compare this to the conventional observational estimate.

* Report the associations between genetic variant and exposure and between genetic variant and outcome, preferably on an interpretable scale.

* Report MR estimates of the relationship between exposure and outcome and the measures of uncertainty from the MR analysis, on an interpretable scale, such as odds ratio or relative risk per SD difference.

* If relevant, please consider translating estimates of relative risk into absolute risk for a meaningful time period.

* Please consider including plots to visualize results (e.g., forest plot, scatterplot of associations between genetic variants and outcome vs between genetic variants and exposure).

---

## [Decision Letter · Decision Letter 2]

19 Dec 2025

Dear Dr. Johansson,

Thank you very much for re-submitting your manuscript "Obesity and renal cancer etiology: a systematic assessment of potential mediators" (PMEDICINE-D-25-02391R2) for review by PLOS Medicine.

Thank you for your detailed response to the reviewers' and editors’ comments. I have discussed the paper with my colleagues and the academic editor with relevant expertise, and it has also been seen again by all three original reviewers. The changes made to the paper were satisfactory to the reviewers. As such, we intend to accept the paper for publication, pending your attention to the editors' comments below in a further revision. When submitting your revised paper, please once again include a detailed point-by-point response to the editorial comments. The remaining issues that need to be addressed are listed at the end of this email.

In revising the manuscript for further consideration here, please ensure you address the specific points made by each reviewer and the editors. In your rebuttal letter you should indicate your response to the reviewers' and editors' comments and the changes you have made in the manuscript. Please submit a clean version of the paper as the main article file. A version with changes marked must also be uploaded as a marked up manuscript file. Please also check the guidelines for revised papers at http://journals.plos.org/plosmedicine/s/revising-your-manuscript for any that apply to your paper.

We ask that you submit your revision by Jan 09 2026. However, if this deadline is not feasible, please contact me or the journal staff (plosmedicine@plos.org) by email, and we can discuss a suitable alternative.

Due to the upcoming holiday season, the journal will operate at reduced capacity from December 22 to January 2. This may cause delays in the manuscript process. We appreciate your understanding.

We look forward to receiving the revised manuscript.

Sincerely,

Alexandra Tosun, PhD

Senior Editor

PLOS Medicine

plosmedicine.org

Comments from Academic Editor:

I think the authors did an excellent job with the revision and the study is an excellent example of a mechanistic study.

Comments from Reviewers:

Reviewer #1: I would like to thank the authors for responding to my comments in detail.

Reviewer #2: The author has addressed all comments.

Reviewer #3: I thank the authors for their response to my comments, which have been addressed satisfactorily.

Requests from Editors:

GENERAL

* Please confirm that your title complies with to PLOS Medicine's style. Your title must be nondeclarative and not a question. It should begin with main concept if possible. "Effect of" should be used only if causality can be inferred, i.e., for an RCT. Please place the study design ("A randomized controlled trial," "A retrospective study," "A modelling study," etc.) in the subtitle (ie, after a colon).

* Statistical reporting: Please revise throughout the manuscript, including tables and figures.

- Please report statistical information as follows to improve clarity for the reader ""22% (95% CI [13,28]; p</=)"".

- Please separate upper and lower bounds with commas instead of hyphens as the latter can be confused with reporting of negative values.

- Please repeat statistical definitions (HR, CI etc.) for each set of parentheses.

* Please ensure that all abbreviations are defined at first use throughout the text (including statistical abbreviations).

* Please ensure that tables and figures, including those in supplementary files, are appropriately referenced in the main text.

* Please review your text for claims of novelty or primacy (e.g. 'for the first time' or ‘novel’) and remove this language.

* Please confirm that any use of statistical terms (such as trend or significant) are supported by the data, and if not please remove them. The term trend should be used only when the test for trend has been conducted.

* Please define all acronyms used in each figure or table in the corresponding legend.

* Please confirm that you used patient-centered language. Please note that patient-centered language is constructed with the use of post-modified nouns putting the person first in the sentence structure.

* Please include the statement on code availability in the data availability statement in the online submission form.

* Please add this statement to the manuscript's Competing Interests: "GDS is an Academic Editor on PLOS Medicine's editorial board."

* Thank you for agreeing to make your data available. At this time, please provide the link to the data repository and accession numbers required for access.

ABSTRACT

* Please confirm that your abstract complies with our requirements, including providing all the information relevant to this study type https://journals.plos.org/plosmedicine/s/submission-guidelines#loc-abstract

* Please confirm that all numbers presented in the abstract are present and identical to numbers presented in the main manuscript text.

* In the abstract, please include the important dependent variables that are adjusted for in the analyses.

METHODS AND RESULTS

* The terms gender and sex are not interchangeable (as discussed in https://www.who.int/health-topics/gender#tab=tab_1 ); please use the appropriate term.

* “We selected each set of genetic instruments according to the gold standard for Mendelian Randomization analyses, previously described (Detailed methods in S1 Appendix) [29].” – Please provide a summary of the methods with sufficient detail in the main text.

* “As a sensitivity analysis, for sex-hormones in females, we adjusted our models in the observational analysis with hormone-replacement therapy and menopausal status which did not alter the results.” – please provide a reference to the relevant table/figure.

* “When considering individual risk factors as mediators, we estimated that insulin mediates between 67% and 22% of the effect of elevated BMI on renal cancer risk based on the cohort and genetic analyses, respectively (Figure 4).” – please explain where 67% and 22% come from? Are these numbers available outside the figure? If so, please ensure to add a relevant reference.

* Please confirm that you specified the variables controlled for in all relevant Tables/Figures.

General Editorial Requests

---

## [Editor Report · Decision Letter 3]

13 Jan 2026

Dear Dr Johansson,

On behalf of my colleagues and the Academic Editor, Steven C Moore, I am pleased to inform you that we have agreed to publish your manuscript "Systematic assessment of obesity-related risk factors in renal cancer etiology: An observational study using both population cohorts and genetic studies" (PMEDICINE-D-25-02391R3) in PLOS Medicine.

I appreciate your thorough responses to the reviewers' and editors' comments throughout the editorial process. We look forward to publishing your manuscript, and editorially there are only two remaining points that should be addressed prior to publication. We will carefully check whether the changes have been made. If you have any questions or concerns regarding these final requests, please feel free to contact me at atosun@plos.org.

Please see below the minor points that we request you respond to:

* Title: We suggest changing the title to: Systematic assessment of obesity-related risk factors in renal cancer etiology: A mediation analysis using cohort data and Mendelian randomization

* Thank you for including the details of the ethical approvals for the UK Biobank and the Västerbotten Intervention Study. Please confirm that your study did not require separate ethical approval.

Before your manuscript can be formally accepted you will need to complete some formatting changes, which you will receive in a follow up email (including the editorial requests above). Please be aware that it may take several days for you to receive this email; during this time no action is required by you. Once you have received these formatting requests, please note that your manuscript will not be scheduled for publication until you have made the required changes.

PRESS

Sincerely,

Alexandra Tosun, PhD

Senior Editor

PLOS Medicine